# High Myopia and Its Associated Factors in JPHC-NEXT Eye Study: A Cross-Sectional Observational Study

**DOI:** 10.3390/jcm8111788

**Published:** 2019-10-25

**Authors:** Kiwako Mori, Toshihide Kurihara, Miki Uchino, Hidemasa Torii, Motoko Kawashima, Mariko Sasaki, Yoko Ozawa, Kazumasa Yamagishi, Hiroyasu Iso, Norie Sawada, Shoichiro Tsugane, Kenya Yuki, Kazuo Tsubota

**Affiliations:** 1Department of Ophthalmology, Keio University School of Medicine, 35 Shinanomachi, Shinjuku-ku, Tokyo 160-8582, Japan; morikiwako@gmail.com (K.M.); uchinomiki@yahoo.co.jp (M.U.); htorii@2004.jukuin.keio.ac.jp (H.T.); motoko326@gmail.com (M.K.); mariko.sasaki@a2.keio.jp (M.S.); ozawa@a5.keio.jp (Y.O.); yukikenya114@gmail.com (K.Y.); tsubota@z3.keio.jp (K.T.); 2Laboratory of Photobiology, Keio University School of Medicine, 35 Shinanomachi, Shinjuku-ku, Tokyo 160-8582, Japan; 3Department of Public Health Medicine, Faculty of Medicine, and Health Services Research and Development Center, University of Tsukuba, 1-1-1 Tennodai, Tsukuba, Ibaraki 305-8575, Japan; 4Public Health, Department of Social Medicine, Osaka University, Graduate School of Medicine, Osaka University Graduate School of Medicine, 2-2 Yamadaoka, Suita, Osaka 565-0871, Japan; iso@pbhel.med.osaka-u.ac.jp; 5Epidemiology and Prevention Group, Center for Public Health Sciences, National Cancer Center, 5-1-1 Tsukiji, Chuo-ku, Tokyo 104-0045, Japan; nsawada@ncc.go.jp (N.S.); stsugane@ncc.go.jp (S.T.)

**Keywords:** high myopia, intraocular pressure, associated factors

## Abstract

The increasing prevalence of high myopia has been noted. We investigated the epidemiological characteristics and the related factors of high myopia in a Japanese adult population. Japan Public Health Center-Based Prospective Study for the Next Generation (JPHC-NEXT) Eye Study was performed in Chikusei-city, a rural area in mid-east Japan, between 2013 and 2015. A cross-sectional observational analysis was conducted to investigate prevalence and related factors of high myopia. A total of 6101 participants aged ≥40 years without a history of ocular surgeries was included. High myopia was defined as a spherical equivalent refraction of ≤−6.00 diopters according to the American Academy of Ophthalmology. Potential high myopia-related factors included intraocular pressure (IOP), corneal structure, corneal endothelial cell density, age, height, body mass index, heart rate, blood pressure, biochemical profile, and current history of systemic and ocular disorders. The odds ratios of high myopia were estimated using the logistic regression models adjusted for the associated factors. The prevalence of high myopia was 3.8% in males and 5.9% in females with a significant difference. Age was inversely associated, IOP was positively associated, and none of other factors were associated with high myopia in both sexes. In conclusion, only age and IOP were associated with high myopia in this community-based sample.

## 1. Introduction

Myopia is one of the most prevalent conditions of the eye. It causes visual impairment in both children and adults that is usually correctable by optical aids such as glasses and contact lenses. High myopia is generally determined as −6.00 diopters (D) or less in refraction and axial lengths of 26.5 mm and more by the American Academy of Ophthalmology [1]. High myopia is associated with progressive and excessive elongation of the eyeball, which results in various funduscopic changes in the posterior fundus, and increases the risk of pathologic myopia, which may cause irreversible vision loss such as glaucoma, retinal detachment, and macular degeneration [2]. High myopia is a major cause of blindness in many countries [3], and the prevalence of myopia and high myopia is expected to increase globally from 2000 to 2050 [4]. Thus, it is important to manage myopia progression and to prevent myopia-related ocular complications and vision loss the approximately 1 billion people with high myopia [4].

Over the past few decades, some studies have provided information on the prevalence and risk factors for myopia, including genetic predisposition and environmental factors such as extended near work, less exercise, and luck of outdoor activities [5,6,7]. However, the etiology of myopia remains unclear. The length of indoor or outdoor activities [8,9,10], ethnicity [11], vitamins [12], diabetes [13], reading habits [14], body stature, lifestyles, and light environment [15,16,17] were suggested as associated factors for the progression of myopia.

Human lifestyles are rapidly changing. The factors that may affect the progression of myopia are essential to understanding and to finding countermeasures for myopia. Community-based population-based research is a reliable way to elucidate the associated factors. In order to elicit the high-myopia-associated factors, it was thought to be important to eliminate biases and prejudices as much as possible. We established this study to screen all the possible factors which were available in the collected data and personal information without manipulation. In addition, this study targeted adult population as subjects. Pathological myopia, which is usually preceded by high myopia and seen mostly in adults, is theoretically thought to be originated from myopia in children. To know possible associated factors for high myopia in adults is considered to be a crucial key to render solutions to diminish the number of future high or pathological myopia in children.

We conducted the Japan Public Health Center-Based Prospective Study for the Next Generation (JPHC-NEXT) Eye Study, an ancillary study of the JPHC-NEXT, to examine the prevalence of refractive status of the participants and factors associated with high myopia in Chikusei, a rural city in mid-east Japan. To our knowledge, this is the first large, community-based study to determine the factors associated with high myopia.

## 2. Materials and Methods

### 2.1. Ethics Approval and Consent to Participate

This study followed the tenets of the World Medical Association’s Declaration of Helsinki. The study protocol was approved by the Institutional Review Boards of Keio University, Osaka University, the University of Tsukuba, and the National Cancer Center. Written informed consent was obtained from all the participants.

### 2.2. Study Design and Participants

JPHC-NEXT Eye Study was performed in two regions: Saku-city and Chikusei-city. The subjects of this study are participants of annual checkups who are aged 40 years or more in Chikusei-city, and a total of 7098 subject who had taken ocular examination between 2013 and 2015 were included in this study. Out of them, 997 participants who had undergone ocular surgeries were excluded because their refraction may have changed after the procedures. In the end, 6101 participants aged from 40 to 93 years (86.0%) were included in this analysis (Figure 1).

### 2.3. Screening Examination

The screening included ophthalmic examinations (refraction, intraocular pressure, central corneal thickness, and corneal endothelium density); measurement of height (HT); weight (WT); blood pressure; and serum laboratory data including glutamic oxaloacetic transaminase (GOT), glutamic pyruvic transaminase (GPT), gamma-glutamyl transpeptidase (GGTP), total cholesterol (T cholesterol), fasting triglyceride (triglycerides), high density lipoprotein (HDL cholesterol), low density lipoprotein (LDL cholesterol), fasting blood glucose (glucose), hemoglobin A1c (HbA1c), and creatinine; and a history of systemic and ocular disorders. Serum T cholesterol, triglycerides, HDL cholesterol, LDL cholesterol, HbA1c, GGTP, and creatinine were measured by visible absorption spectrometry. Serum glucose, GOT, and GPT were measured using ultraviolet absorption spectrometry. Blood pressure was measured two times, and the second reading was adopted. Height and body weight were measured with AD-6350 (A&D Company Ltd., Tokyo, Japan). For blood pressure, TN2657P, TM 2657P, TM 2655P, and UM-102 (A&D Company Ltd., Tokyo, Japan) were used; for biochemical examinations, BM8060G (NDK Inc., Sagamihara, Japan), LUMIPULSE^®^ G1200 (FUJIREBIO Inc., Tokyo, Japan) and AIA2000 (Tosoh Corp., Tokyo, Japan) were used; and for blood glucose tests, BM9130 (NDK Inc., Sagamihara, Japan) was used. Refractive status and intraocular pressure (IOP) were measured using an auto refractometer (Tonoref II, Nidek, Gamagori, Japan). Central corneal thickness and corneal endothelium density were measured by using a specular-type pachymeter (Specular microscope XIII, Konan, Nishinomiya, Japan).

### 2.4. Information on Past Medical History

Information on histories of hypertension, diabetes, and dyslipidemia was collected through face-to-face interviews at the baseline survey. Likewise, inquiry about smoking history and alcohol history as well as histories of ocular disease and its surgery were also performed.

### 2.5. Definitive Examination

The spherical equivalent refraction (SEq) was calculated from the refraction using the following formula: the full spherical power plus half of the cylindrical power. Initially, the mean values of the SEq of the right eyes and the left eyes were compared in the entire population of the current study. There was no significant difference in the median SEq between the right eyes and the left eyes (−0.13 vs −0.00 (D), *p* = 0.10). Thus, we used the SEq of the right eyes to evaluate the refractive status in this study. A high myopia was defined as SEq of −6.00 D or less based on the American Academy of Ophthalmology criteria [1].

### 2.6. Statistical Analysis

The prevalence of high myopia was calculated in total participants, in men and women, and in the groups classified by age. Chi-square test for the prevalence of high/non-high myopia in each sex was performed. When it demonstrated significant difference, comparison of variables between high myopia and non-high myopia groups in each sex was performed; continuous variables showing parametric distribution were analyzed with Student’s t-test, and variables showing nonparametric distribution were analyzed with Mann–Whitney U test. Logistic regression was used to estimate odds ratios (ORs) and 95% confidence intervals (CIs), adjusting for age and for likely possible risk factors associated with high myopia. Multivariate logistic analyses were performed using variables showing significant difference in each sex as adjustment factors, since variables showing significance (*p* < 0.05) are different from each other by comparison between non-high myopia and high myopia groups in both sexes. Units are different among variables, and therefore, they are categorized by quartiles. A *p* value of less than 0.05 was considered statistically significant. Statistical analyses were performed with the SPSS version 23.0 for Windows (IBM, Armonk, NY, USA).

## 3. Results

### 3.1. Participants

Among a total of 5987 subjects, the numbers of men and women were 2427 and 3557, respectively. The prevalence of high myopia in the current study was 5.0%, 3.8% in men and 5.9% in women. High myopia was more prevalent among women than men (*p* < 0.001). According to the age distribution, the prevalence of high myopia in the 40s, 50s, 60s, 70s, 80s, and 90s was 10.6%, 8.8%, 3.5%, 2.0%, 1.2%, and 0.0%, respectively (Figure 2). The median SEq of the right eyes in total was −0.13 D, ranging from −23.13 D to +10.25 D, with the 1st quartile at −1.38 and the 3rd quartile at +0.88 D. In the men, the median SEq was −0.13 D, ranging from −18.88 D to +10.25 D, with the 1st quartile at −1.25 and the 3rd quartile at +0.88 D. In the women, the median SEq was −0.13 D, ranging from −23.13 D to +6.88 D, with the 1st quartile at −1.63 and the 3rd quartile at +0.88 D. The median SEq of the right eye was not significantly different between genders (*p* = 0.52).

### 3.2. Associations Between Systemic Factors and High Myopia

We evaluated the associations between systemic factors and high myopia (Table 1). The mean age for the high myopia group was significantly lower than the non-high myopia group (55.7 ± 10.5 vs 62.8 ± 10.3 years, *p* < 0.001). The peak of the high myopia prevalence by age was from 40 to 49 years (6.2% among the men and 13.0% among the women). The mean height for the high myopia group was significantly higher than others (158.8 ± 8.0 vs 157.5 ± 8.7 cm, *p* = 0.009). A similar difference was also found in the age and heights of men and women, respectively (Table 1). No significant difference between the high myopia group and non-high myopia group was observed in weight, waist, and diastolic blood pressure. Among the laboratory data, T cholesterol, triglycerides, and LDL cholesterol were significantly different between the two groups in men. Glucose and HbA1c were significantly different between the two groups in the women. Regarding the relationship between alcohol intake and high myopia, ORs for high myopia and alcohol intake history were 1.166 (95%CI 0.371–3.662) in the men and 0.900 (95%CI 0.467–1.732) in the women. ORs for high myopia and tobacco use history were 0.598 (95%CI 0.340–1.051) in the men and 0.967 (95%CI 0.593–1.577) in the women (Appendix A).

### 3.3. Associated Between Ocular Features and High Myopia

Associations between ocular features and high myopia were also evaluated (Table 1). The median IOP was significantly higher for the high myopia group than the others (14.30 vs 13.30 mmHg, *p* < 0.001). Similar differences were found in the median IOPs of the men or women, respectively. A significant difference was also observed in the median corneal radius between the high myopia group and the others (7.62 vs. 7.64 mm, *p* = 0.024). However, no significant difference was found in the median corneal radii of the men and women, respectively. There was no significant difference in central corneal thickness and corneal endothelial cell number between the two groups.

### 3.4. IOP and Age Associated with High Myopia

A multivariable logistic regression analysis was performed to identify the factors associated with high myopia (Table 2 and Appendix A). The ORs adjusted for multivariable factors were significant for age and IOP in the men and the women. In men, the values were as follows: the ORs adjusted with other factors for high myopia in each age group were 40–58 years (OR 1), 59–65years (OR 0.44, 95%CI 0.22–0.89), 66–70 years (OR 0.43, 95%CI 0.20–0.93), and no younger than 71 years (OR 0.48, 95%CI 0.22–1.06, *p* for trend 0.049). The ORs adjusted with other factors for high myopia in the groups of IOP were ≤11.2 mmHg (OR 1), 11.3–13.2 mmHg (OR 3.34, 95%CI 0.93–12.00), 13.3–15.2 mmHg (OR 5.18, 95%CI 1.49–18.04), and ≥15.3 mmHg (OR 7.73, 95%CI 2.24–26.77), respectively (*p* for trend <0.001).

In women, the values were as follows: the ORs adjusted with other factors for high myopia in each age group were 40–54 years (OR 1), 55–62 years (OR 0.57, 95%CI 0.36–0.90), 63–68 years (OR 0.35, 95%CI 0.21–0.59), and no younger than 69 years (OR 0.21, 95%CI 0.11–0.41) (*p* for trend <0.001). The ORs adjusted with other factors in each IOP group were ≤11.6 mmHg (OR 1), 11.7–13.6 mmHg (OR 2.23, 95%CI 1.23–4.05), 13.7–15.6 mmHg (OR 2.25, 95%CI 1.21–4.16), and ≥15.7 mmHg (OR 2.33, 95%CI 1.24–4.37), respectively (*p* for trend 0.023). In summary, as the age increases, the OR decreases and, as the IOP increases, the OR increases in both men and women.

## 4. Discussion

This community-based study showed the prevalence of high myopia and associated factors including physical, ocular, and demographic factors among adult Japanese for the first time. The factors associated with myopia were widely investigated as we analyzed biochemistry tests, blood pressure, height, body weight, habit of smoking, alcohol intake, past medical history, and present diseases. In this current study, we found that high myopia is more prevalent in women and younger age and that it has higher IOP.

High myopia affects approximately 1–4% of adults aged ≥40 years, and its prevalence was higher in some studies of East Asian adults and adolescents [18,19,20,21,22,23,24,25]. Our findings showed that the prevalence of high myopia was 5.0%, which was no less than the generally affected rates, although Chikusei-city is in a rural area where its prevalence has usually been lower compared to urban areas [25]. In our study, the high myopia rate in the older population was relatively low and the younger generation had a higher prevalence of high myopia, which may reflect cohort effects. Although the reason for this cohort effects is unknown, the prevalence of high myopia could be expected to increase in the future.

Some previous studies reported the relationship of height [26,27,28] and BMI [29,30,31] with myopia; however, we did not find such association. A possible reason for this discrepancy is confounding; the previous reports did not adjust for age and any other confounding factors. In fact, BMI and height were shown to be associated with high myopia in unadjusted models and adjustment for age resulted in the elimination of this significance in our study, presumably because age is a strong predisposing factor for BMI and height reflecting cohort effects.

As for the other laboratory factors, there was no significant difference. A few studies suggested that hyperglycemia and hyperlipidemia led to myopic shift, whereas other studies revealed that the refractive shift was more likely hyperopic with hypoglycemia [32,33,34,35]. Further analysis is needed to elucidate the influence of metabolic shift.

In terms of alcohol intake and smoking history, both factors did not show any associations with high myopia according to our results. Also, liver functions, represented by GOT, GPT, and GGTP, did not show any relationship to high myopia. Previous reports also found that there were no significant trends observed between smoking and refractive errors [36].

The percentages of high myopia in men and women were 3.8% and 5.9%, respectively, indicating significant gender difference. It has been reported that female sex had a predisposition of high myopia [37]. Likewise, female sex was proven to have high risk for myopic complications usually caused by high myopia [38]. Hyman et al. reported that female sex is independently associated with faster myopic progression [39], although there have been no reports describing causal relationship between gender and high myopia.

The mean IOP linearly increased parallel to the myopic progression. It has been reported that IOP was associated with central corneal thickness, age, and blood pressure [40,41]. Even after adjustment for these factors, we still found that high IOP was significantly associated with high myopia, while there are conflicting evidences regarding relationship of high myopia and IOP [42,43,44].

Although this cross-sectional study is to investigate the risk factors for high myopia through a large-scale cross-sectional study, a few studies with relatively small number of subjects were undertaken. Mo et al. [37] concluded in their study performed in a rural area in China with 167 participants that females may be a risk factor for myopia and that advanced age is a factor for decreased visual acuity. As for the results obtained from this large-scale study showing that being young, being a woman, and having high IOP are the risk factors for high myopia, we investigated previous articles regarding these factors. Matamoros et al. [45] reported in their study that prevalence of high myopia was higher in 20- to 39-year-old subjects as well as among women than men, which are consistent with our results. Otherwise, being a woman has well been reported to be a risk factor for myopia in some other studies [46]. Attebo et al. [46] also confirmed an age-related increase in hyperopia associated with an age-related decrease in myopia; however, a distinct association between high myopia and age has not been elucidated. Concerning relationships between high myopia and glaucoma, Chen et al. [47] reported high myopia as a risk factor in primary open angle glaucoma (POAG) and elevated intraocular pressure as a well-known major risk factor for POAG. Nevertheless, there have been no reports describing direct association between high IOP and high myopia.

There are some limitations in this study. First of all, the study design is not completely reasonable because the data used for the analysis are practically limited. Although the JPHC-NEXT study is large scale in terms of the participant number, it is basically a universal data collection based on regular checkups performed in a rural area. Therefore, this is not conducted specifically for ophthalmological analysis. The study would have been more informative if it contained questionnaires or interviews concerning the time for near-work and indoor/outdoor activities as well as academic levels and heredity of myopia. In addition, another limitation of this study is the lack of ophthalmological analysis including the measurement of axial length and lens corrections, which are crucial to determine the participants’ classification of non-/high myopia. This study should be completed with these detailed data to make the conclusions more reliable; however, it could be done only beyond the limitation of this cross-sectional study. Owing to the large-scale study, instead, we were able to effectively assess a large number of basically healthy participants altogether and to draw these critical results. The factors associated with high myopia were undetermined. This study did not suggest that height, BMI, blood glucose, hypocholesteremia, liver dysfunction, kidney dysfunction, smoking, and alcohol intake were associated with high myopia, whereas women, young age, and high IOP were found to be related to high myopia. Meanwhile, high IOP and young age were found to be risk factors for high myopia, which may indicate the path for future studies concerning myopia control.

## 5. Conclusions

In conclusion, this epidemiological study performed in a Japanese rural area revealed significant results: being a women, being young, and having high IOP were factors associated with high myopia, while height, BMI, cholesterol level, glucose level, and other considered possible risk factors for high myopia in previous studies were not associated with prevalence of myopia.

## Figures and Tables

**Figure 1 jcm-08-01788-f001:**
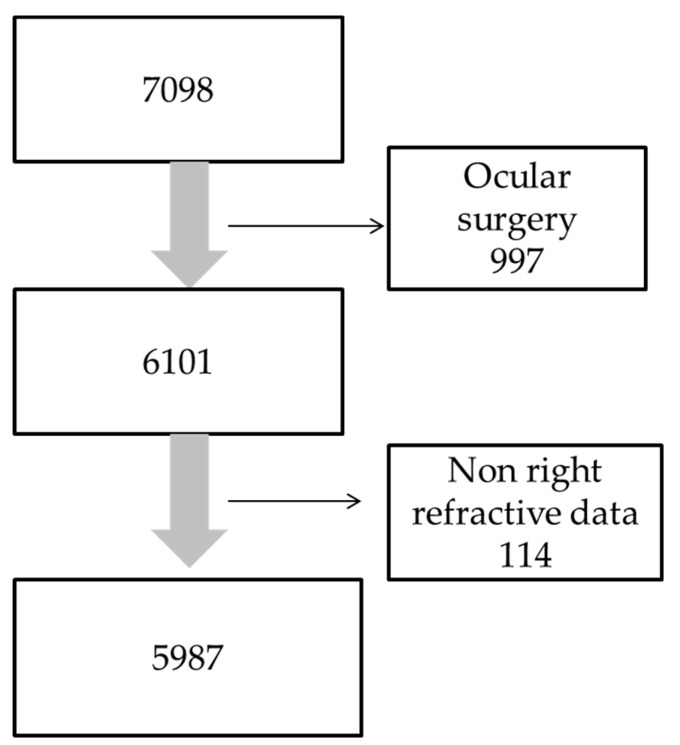
Study flow chart of this study. The number of participants who were 40 years old and over was 7098. Of them, 5987 participants were defined as the subjects after excluding 997 participants who had a history of ocular surgery and 114 participants who did not have refractive indices of their right eye.

**Figure 2 jcm-08-01788-f002:**
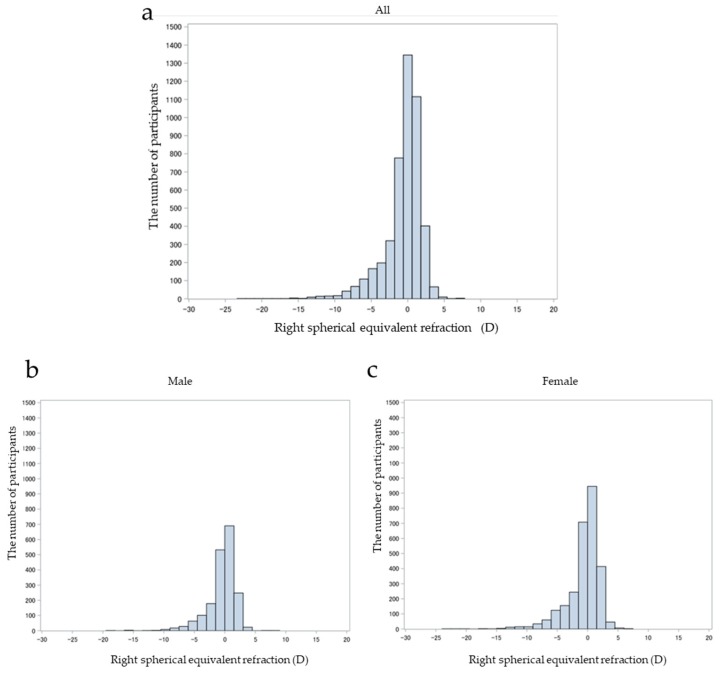
Distribution of spherical equivalent refraction in Chikusei-city: (**a**) the distribution map shows the spherical equivalent refraction of the right eye in the subjects, (**b**) spherical equivalent refraction of the right eye in men subjects, and (**c**) spherical equivalent refraction of the right eye in women subjects.

**Table 1 jcm-08-01788-t001:** The association between systemic and ocular features and high myopia.

	Mean ± SD		
Variables	Non-High Myopia	High Myopia	*p* Value
All (*n* = 5984)			
Age	62.8 ± 10.3	55.7 ± 10.5	<0.001
HT	157.6 ± 8.7	158.8 ± 8.0	0.009
BMI	23.2 ± 3.3	22.8 ± 4.1	0.093
HR	63.8 ± 13.5	64.3 ± 14.8	0.669
SBP	124.8 ± 17.6	121.0 ± 18.9	0.001
DBP	74.4 ± 11.6	73.9 ± 13.3	0.504
GOT	24.1 ± 37.9	22.1 ± 8.4	0.422
GPT	21.3 ± 31.7	20.6 ± 12.9	0.737
GGTP	34.0 ± 43.9	30.0 ± 24.3	0.022
T CHOL	209.4 ± 35.6	213.0 ± 35.0	0.082
TG	96.0 ^†^	84.0 ^†^	0.595 ^††^
HDL CHOL	63.1 ± 15.7	65.3 ± 15.4	0.034
LDL CHOL	126.7 ± 31.7	130.8 ± 31.7	0.052
GLU	96.0^†^	93.0 ^†^	<0.001 ^††^
HbA1c	5.8 ± 0.7	5.7 ± 0.5	0.004
Creatinine	0.7 ± 0.3	0.7 ± 0.1	0.014
IOP	13.3 ^†^	14.3 ^†^	<0.001 ^††^
Corneal radius	7.6 ^†^	7.6 ^†^	0.070 ^††^
Central corneal thickness	553.0 ^†^	557.0 ^†^	0.102 ^††^
Corneal endothelial cell number	2747.0 ^†^	2740.0^†^	0.174 ^††^
Men (*n* = 2427)			
Age	64.2 ± 10.4	58.7 ± 11.2	<0.001
HT	164.8 ± 6.5	166.7 ± 6.4	0.007
BMI	23.8 ± 3.1	24.3 ± 4.2	0.169
HR	62.6 ± 13.7	63.4 ± 14.5	0.666
SBP	128.6 ± 17.1	129.4 ± 17.9	0.684
DBP	76.8 ± 11.6	78.5 ± 12.4	0.176
GOT	26.4 ± 58.1	23.6 ± 8.0	0.677
GPT	24.7 ± 47.3	24.9 ± 12.1	0.972
GGTP	46.3 ± 58.7	40.0 ± 24.2	0.374
T CHOL	119.3 ± 33.7	209.3 ± 31.3	0.003
TG	105.0 ^†^	117.0 ^†^	0.023 ^††^
HDL CHOL	57.0 ± 14.5	56.4 ± 13.7	0.733
LDL CHOL	121.0 ± 30.7	129.7 ± 29.9	0.020
GLU	99.0 ^†^	99.0 ^†^	0.597 ^††^
HbA1c	5.8 ± 0.8	5.7 ± 0.5	0.215
Creatinine	0.9 ± 0.3	0.9 ± 0.1	0.733
IOP	13.3 ^†^	15.0 ^†^	<0.001 ^††^
Corneal radius	7.7 ^†^	7.7 ^†^	0.198 ^††^
Central corneal thickness	556.0 ^†^	565.0 ^†^	0.391 ^††^
Corneal endothelial cell number	2793.0 ^†^	2789.5 ^†^	0.538 ^††^
Women (*n* = 3557)			
Age	61.8 ± 10.2	54.4 ± 9.9	<0.001
HT	152.5 ± 6.2	155.4 ± 5.9	0.001
BMI	22.8 ± 3.5	22.2 ± 3.9	0.012
HR	64.6 ± 13.4	64.6 ± 15.0	0.964
SBP	122.1 ± 17.5	117.4 ± 18.2	<0.001
DBP	72.7 ± 11.3	71.9 ± 13.2	0.358
GOT	22.5 ± 9.2	21.5 ± 8.5	0.168
GPT	19.0 ± 11.7	18.7 ± 12.9	0.815
GGTP	25.4 ± 26.2	25.6 ± 23.2	0.920
T CHOL	216.4 ± 35.1	214.6 ± 36.4	0.461
TG	90.0 ^†^	83.0 ^†^	0.076 ^††^
HDL CHOL	67.3 ± 15.1	69.2 ± 14.4	0.117
LDL CHOL	130.6 ± 31.8	131.3 ± 32.5	0.783
GLU	95.0 ^†^	91.0 ^†^	<0.001 ^††^
HbA1c	5.8 ± 0.6	5.6 ± 0.5	0.011
Creatinine	0.6 ± 0.1	0.6 ± 0.1	0.235
IOP	13.7 ^†^	14.0 ^†^	0.038 ^††^
Corneal radius	7.6 ^†^	7.6 ^†^	0.327 ^††^
Central corneal thickness	551.0 ^†^	553.5 ^†^	0.159 ^††^
Corneal endothelial cell number	2740.0 ^†^	2725.0 ^†^	0.159 ^††^

Non-high myopia: SEq > −6D, High myopia: SEq ≤ −6D. ^†^: Median. ^††^: Mann–Whitney U test. HT = height, BMI = body mass index, HR = heart rate, SBP = systolic blood pressure, DBP = diastolic blood pressure, GOT = glutamic-oxaloacetic transaminase, GPT = glutamate pyruvate transaminase, GGTP = gamma-glutamyl transpeptidase, T CHOL = total cholesterol, TG = triglyceride, HDL CHOL= high-density lipoprotein cholesterol, LDL CHOL = low-density lipoprotein cholesterol, GLU = glucose, HbA1c = Hemoglobin A1c, IOP = intraocular pressure.

**Table 2 jcm-08-01788-t002:** Multivariate logistic regression analysis to identify factors associated with high myopia.

		OR	95% CI	*p* Value
Men				
Age	≤58	1		
	59–65	0.44	(0.221–0.886)	
	66–70	0.43	(0.197–0.930)	
	≥71	0.48	(0.219–1.059)	
	*p* for trend			0.049
HT	≤160	1		
	161–164	0.95	(0.388–2.318)	
	165–168	1.42	(0.620–3.271)	
	≥169	1.65	(0.722–3.784)	
	*p* for trend			0.131
BMI	≤21.6	1		
	21.7–23.5	1.41	(0.660–3.020)	
	23.6–25.6	0.96	(0.423–2.188)	
	≥25.7	1.07	(0.479–2.368)	
	*p* for trend			0.836
SBP	≤116	1		
	117–127	0.61	(0.272–1.364)	
	128–138	0.98	(0.481–1.978)	
	≥139	0.83	(0.396–1.741)	
	*p* for trend			0.912
TG	≤73	1		
	74–105	2.93	(1.126–7.616)	
	106–149	3.10	(1.183–8.103)	
	≥150	2.79	(1.018–7.619)	
	*p* for trend			0.117
HDL CHOL	≤46	1		
	47–54	0.78	(0.374–1.625)	
	55–64	0.99	(0.473–2.065)	
	≥65	1.14	(0.529–2.442)	
	*p* for trend			0.682
LDL CHOL	≤100	1		
	101–119	2.07	(0.909–4.729)	
	120–139	1.44	(0.615–3.371)	
	≥140	1.72	(0.758–3.892)	
	*p* for trend			0.348
HbA1c	≤5.3	1		
	5.4–5.5	0.84	(0.390–1.797)	
	5.6–5.9	0.75	(0.355–1.579)	
	≥6.0	0.83	(0.377–1.811)	
	*p* for trend			0.453
IOP	≤11.2	1		
	11.3–13.2	3.34	(0.927–12.002)	
	13.3–15.2	5.18	(1.487–18.037)	
	≥15.3	7.73	(2.235–26.768)	
	*p* for trend			<0.001
Central corneal thickness	≤531	1		
	532–556	0.93	(0.421–2.053)	
	557–583	1.05	(0.483–2.269)	
	≥584	1.01	(0.471–2.180)	
	*p* for trend			0.847
Women				
Age	≤54	1		
	55–62	0.57	(0.358–0.904)	
	63–68	0.35	(0.208–0.586)	
	≥69	0.21	(0.106–0.405)	
	*p* for trend			<0.001
HT	≤147	1		
	148–152	0.88	(0.486–1.594)	
	153–156	1.03	(0.571–1.872)	
	≥157	1.35	(0.752–2.424)	
	*p* for trend			0.195
BMI	≤20.3	1		
	20.4–22.3	0.76	(0.481–1.205)	
	22.4–24.6	0.58	(0.344–0.977)	
	≥24.7	0.68	(0.399–1.156)	
	*p* for trend			0.120
SBP	≤108	1		
	109–120	0.74	(0.458–1.191)	
	121–132	0.87	(0.520–1.457)	
	≥133	1.02	(0.605–1.729)	
	*p* for trend			0.698
TG	≤66	1		
	67–89	0.85	(0.530–1.352)	
	90–124	0.77	(0.455–1.288)	
	≥125	0.94	(0.547–1.619)	
	*p* for trend			0.624
HDL CHOL	≤56	1		
	57–65	1.37	(0.810–2.298)	
	66–75	1.25	(0.735–2.116)	
	≥76	1.02	(0.583–1.785)	
	*p* for trend			0.801
LDL CHOL	≤107	1		
	108–127	1.07	(0.629–1.830)	
	128–149	2.07	(1.273–3.352)	
	≥150	1.27	(0.744–2.149)	
	*p* for trend			0.121
HbA1c	≤5.4	1		
	5.5–5.6	1.29	(0.833–1.993)	
	5.7–5.8	0.85	(0.493–1.463)	
	≥5.9	1.03	(0.618–1.731)	
	*p* for trend			0.738
IOP	≤11.6	1		
	11.7–13.6	2.23	(1.226–4.052)	
	13.7–15.6	2.25	(1.213–4.157)	
	≥15.7	2.33	(1.239–4.367)	
	*p* for trend			0.023
Central corneal thickness	≤524	1		
	525–550	0.83	(0.508–1.361)	
	551–574	0.70	(0.423–1.164)	
	≥575	0.92	(0.561–1.523)	
	*p* for trend			0.730

N on-high myopia: SEq > −6D, high myopia: SEq ≤ −6D, OR: Adjusted by age, HT, BMI, SBP, TG, HDL CHOL, LDL CHOL, HbA1c, IOP, and central corneal thickness. HT = height, BMI = body mass index, SBP = systolic blood pressure, TG = triglyceride, HDL CHOL = high-density lipoprotein cholesterol, LDL CHOL= low-density lipoprotein cholesterol, HbA1c = Hemoglobin A1c, IOP = intraocular pressure.

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
