# Peer review of "High Myopia and Its Associated Factors in JPHC-NEXT Eye Study: A Cross-Sectional Observational Study"

_jcm, 2019, doi:10.3390/jcm8111788_

Round 1

Reviewer 1 Report

The authors have endeavored to respond to all the clarifications requested by the reviewers. As much as possible they have answered all of them.
However, I believe that although I understand that they cannot obtain the ophthalmological and questionnaire data (work habits, studies etc.) because it is a longitudinal study that did not have the objective of analyzing only ophthalmological values, in my opinion, this design invalidates the study in itself since serious mistakes could be made in the organization of patient groups and also for omitting very important risk factors in the development of myopia.

Reviewer 2 Report

The responses to the previous review were very well handled.

I have no more suggestions.

This manuscript is a resubmission of an earlier submission. The following is a list of the peer review reports and author responses from that submission.

Round 1

Reviewer 1 Report

The article “High Myopia and its associated factors in JPHC-NEXT Eye Study: a cross-sectional observational study”, is an article of interest, easy to read, well written and with a simple but correct analysis. The total number of participants is very high, more than 6000 participants and many of the analyzed  biochemical factors are novel (BMI, HR, GOT, Creatinine ...) and no association had been established so far with High Myopia, a disease that is taking a lot scientific interest for its increase in prevalence.

However, I believe that in a study with so many participants, from which great conclusions can be drawn, there are serious shortcomings in the design and obtaining data of special variables related with high myopia.. In the face-to-face interview with the participants they should have included the questions of the times in near work, indoors and outdoors activities as well as the level of studies, the time reading and/or with computers or other electronic devices. These are basic results to complete the analysis of the rest of the novel factors studied (and that have not shown any association). In addition, and certainly most importantly, the ophthalmological analysis is very poor and important factors have not been taken into account such as the measurement of axial length in patients, lens correction and the possible ocular surgeries of participants. These variables are essential to determine if the participants are from the high myopia group (> 26.5 mm) or not. The lack of these determinations can lead to false negatives and consequently in erroneous conclusions. It is a serious mistake and although they point it out in the limitations, they do not explain it with sufficient forcefulness and reasoning, since the analysis and conclusions of the study remain very depleted. They have to complete the study with these data to make the conclusions more reliable, it is essential.

The conclusions resulting from the study are interesting and well obtained with the analyzes they have performed. The greatest risk for being a woman, young and with IOP, is an interesting contribution, although not new, since they are factors already known and established in other previous studies, the biochemical analysis has not contributed any factor associated with High myopia in this Japanese rural population.

At the structural level, there are certain errors to solve before publication.

Material and methods: they need to complete with model and commercial company of the several devices that have been used to measure many of the study variables (height, weight, serum laboratory data, blood pressure).

Results: The results of table 1 are divided into men and women but there are no unified results in the table, however in the text they mainly talk about these results, there is no concordance. Table 2 is too large, only the positive or most important results should appear, for the rest of data is better use the supplementary material.

Discussion: The reason for the lack of such important variables as axial length, near work time, outdoor activities, lens examinations is very little explained. I think it is necessary to give a convincing explanation of the lack of these data. The previous studies on the risk of High Myopia with the increase of IOP should be explained more, there are studies of the relationship of High Myopia and glaucoma. This point should be discussed further.

Reviewer 2 Report

This paper reports the risk factors associated with high myopia in the JPHC-NEXT EYE-Study. The risk factors have been  examined with a cross-sectional observational study

The results are interesting but this study can be be improved notably for the statistical part.

In the Materials and Methods,

The paragraph line 73 to line 76 is repeated line 83 to line 87.

In the statistical analyses part:

Line 122, a sentence about the descriptive analyses is missing. It would be precised that description and tests have been performed in the total population. The choice to present separately the results for the men and the women has to be justified. An interaction test has been performed ? There is a problem of multiple testing in this study. It is well known that the more tests you run, the greater the likelihood of a chance finding. In addition the variables must be presented in the same format in all analyzes. Thus, it will be better to present the table 1 without P value, only for the descriptive part. The selection of the variables to be included in the multivariate model can be done directly from the age-adjusted logistic models. Lines 129-131, theses variables have not to be listed in the method section. This is a result. In addition, the variables significantly associated with myopia for men and for women are not the same. It will be better to include, in the multivariate models, the variables associated with myopia in age-adjusted logistic models according to gender (a specific model for men and another for women). Line 129, after « regression models » the dot is missing. Line 129, please, precise the P-value chosen for the selection of the variables to be included in the mulivariate models.

For the results

- The description of the population (number of total subjects, number of men, number of women….) is missing in the text and in the tables.

For the discussion

- Comparison of these results with other studies should be further detailed.

- For the comparison with other studies, it will be interesting to have information on mean age, area (urban, rural), country… and on the format of variables used (example for BMI, have the studies used quartiles ? threshold ? continuous variable ?)

- Line 216, the results of the studies on children are not really comparable to those on adults. Thus, the reference 15, line 216 has to be deleted.

- Paragraph Limitations (lines 241-247), the major limit of this study is that the multivariate models have not been adjusted on heredity of myopia (at least one parent with myopia). This is a main factor. This limit should be reported first. The results could be different if this factor had been taken into account in the multivariate models.

For the conclusion

Line 258, the dot is missing.

For the references

The doi is missing for reference 25 (line 341), reference 31 (line 356), reference 33 (line 361), reference 36 (line 368), reference 43 (line 387), reference 44 (line 389)